# Doubtful Clinical Value of Subtyping Anti-U1-RNP Antibodies Regarding the RNP-70 kDa Antigen in Sera of Patients with Systemic Lupus Erythematosus

**DOI:** 10.3390/ijms241210398

**Published:** 2023-06-20

**Authors:** Awais Ahmad, André Brylid, Charlotte Dahle, Muna Saleh, Örjan Dahlström, Helena Enocsson, Christopher Sjöwall

**Affiliations:** 1Department of Biomedical and Clinical Sciences, Division of Inflammation and Infection/Clinical Immunology & Transfusion Medicine, Linköping University, SE-581 85 Linköping, Sweden; awais.ahmad@regionostergotland.se (A.A.); charlotte.dahle@regionostergotland.se (C.D.); 2Department of Biomedical and Clinical Sciences, Division of Inflammation and Infection/Rheumatology, Linköping University, SE-581 85 Linköping, Sweden; muna.saleh@liu.se (M.S.);; 3Department of Behavioural Sciences and Learning, Swedish Institute for Disability Research, Linköping University, SE-581 83 Linköping, Sweden

**Keywords:** autoantibodies, mixed connective tissue disease, primary Sjögren’s syndrome, small nuclear ribonucleoprotein antibodies, systemic lupus erythematosus

## Abstract

The detection of antinuclear antibodies is central to the diagnosis and prognosis of systemic lupus erythematosus (SLE), primary Sjögren’s syndrome (pSS) and mixed connective tissue disease (MCTD). Anti-U1-RNP and anti-RNP70 antibodies were assayed in the sera of patients with SLE (*n* = 114), pSS (*n* = 54) and MCTD (*n* = 12). In the SLE group, 34/114 (30%) were anti-U1-RNP positive, and 21/114 (18%) were both anti-RNP70 positive and anti-U1-RNP positive. In the MCTD group, 10/12 (83%) were anti-U1-RNP positive, and 9/12 (75%) were anti-RNP70 positive. Only one individual with pSS was antibody positive (for both anti-U1-RNP and anti-RNP70). All anti-RNP70-positive samples were also anti-U1-RNP positive. Anti-U1-RNP-positive subjects with SLE were younger (*p* < 0.0001); showed lower concentrations of complement protein 3 (*p* = 0.03); had lower eosinophil (*p* = 0.0005), lymphocyte (*p* = 0.006) and monocyte (*p* = 0.03) counts; and had accrued less organ damage (*p* = 0.006) than the anti-U1-RNP-negative SLE patients. However, we observed no significant clinical or laboratory parameter differences between the anti-U1-RNP-positive individuals with/without anti-RNP70 in the SLE group. In conclusion, anti-RNP70 antibodies are not exclusive to MCTD but are rarely detected in pSS and healthy individuals. In SLE, anti-U1-RNP antibodies are associated with a clinical phenotype that resembles MCTD, with hematologic involvement and less damage accrual. Based on our results, the clinical value of subtyping anti-RNP70 in anti-U1-RNP-positive sera appears to be of limited value.

## 1. Introduction

The analysis of antinuclear antibodies (ANA) is a key investigative tool for patients with suspected systemic inflammatory diseases, although the diagnostic performance of ANA screening depends on the employed techniques [1]. In the clinical routine, ANA testing usually includes the most important autoantibody specificities that have established associations with clinical disease, e.g., autoantibodies directed against double-stranded DNA (dsDNA), the Smith antigen (Sm), small nuclear ribonucleoproteins (snRNPs), topoisomerase I (Scleroderma-70), histidyl-tRNA synthetase (Jo-1), Ro52/TRIM21, Ro60 and La [2,3].

The snRNP group of antigens includes U1-RNP, U2-RNP, U4/U6-RNP and U5-RNP. Antibodies against U1-RNP are relevant clinically and required for the diagnosis of mixed connective tissue disease (MCTD), although they are also frequently found in cases of systemic lupus erythematosus (SLE) [4]. U1-RNP is part of the spliceosome, which removes introns from pre-messenger RNA and consists of U1-RNA bound to the 70 kDa subunit of the U1-snRNP complex (RNP70), RNP-A and RNP-C [5,6]. Together with the Sm antigens, snRNP constitutes a core group, of which RNP70, RNP-A, RNP-C, SmB/B’, SmD1 and SmD3 are the most important autoantigens [6,7]. Cross-reactivities between SmB/B’ and RNP-A or RNP-C have been reported, and autoantibodies against RNP70 and SmD1 or SmD3 are, therefore, regarded as the most clinically relevant [8,9,10].

While antibodies against U1-RNP (anti-U1-RNP) are detected in up to 30% of patients with SLE, the diagnostic specificity is low because these ANAs are also found in other autoimmune diseases [2,4,11]. For diagnosing MCTD, positivity for anti-U1-RNP antibodies is required, and high titers of these antibodies are usually measured at the time of diagnosis [12,13]. Whereas SLE is a multisystem disease that can attack several organs (e.g., the skin, kidneys and joints), the common clinical manifestations of MCTD include puffy hands, synovitis, myositis, Raynaud’s phenomenon (RP) and sclerodactyly [14]. In cases of MCTD, the anti-U1-RNP antibodies are most often directed against RNP70 (anti-RNP70) and are often also directed against RNP-A and RNP-C [15,16,17,18].

Historically, different approaches have been used to detect anti-snRNP antibodies. The highly specific, albeit less sensitive, immunodiffusion and counterimmuno-electrophoresis methods using native purified antigens were standard for decades but have over time been replaced by more sensitive but less specific automated ELISA-based methods using recombinant antigens. In the indirect immunofluorescence (IIF) microscopy of human epithelial-2 (HEp-2) cells, anti-snRNP antibodies typically give a coarse speckled pattern (AC-5) [1], although an antigen-specific test is required to confirm the specificity [19].

There is gap in the knowledge regarding the added value of analyzing anti-RNP70 antibodies in serum samples that are positive for anti-U1-RNP antibodies. Therefore, in this retrospective study, we evaluate two commercial fluoroenzymatic immunoassays (FEIA; EliA™ U1-RNP and EliA™ RNP70) for measuring anti-U1-RNP and anti-RNP70 antibodies in samples from four cohorts: SLE, MCTD, primary Sjögren’s syndrome (pSS) and healthy blood donors (HBDs). Our primary aim was to determine whether antibodies against RNP70 provide any additional clinical information of importance in cases of SLE and should, as a consequence, be included in the routine evaluation of these patients. A secondary aim was to describe the associations between anti-U1-RNP positivity with or without concomitant positivity for anti-RNP70 antibodies and various clinical and laboratory variables.

## 2. Results

### 2.1. Differences within the SLE Group Based on Clinical Routine Analyses of Anti-U1-RNP Antibodies

The samples from patients with confirmed SLE (*n* = 114) consisted of sera that had been judged to be either anti-U1-RNP positive (*n* = 53) or anti-U1-RNP negative (*n* = 61) in the clinical routine (using the addressable laser bead immunoassay (ALBIA) and/or EUROLINE immunoblot) (Figure 1). The anti-U1-RNP-positive patients were younger (*p* < 0.0001), had a shorter duration of SLE (*p* < 0.0001), more often fulfilled the hematologic disorder criteria of the American College of Rheumatology (ACR) (*p* = 0.04) and had an increased frequency of RP (*p* = 0.03). At the time of sampling, there was a non-significant trend of higher disease activity, as assessed using the clinical SLE disease activity index 2000 (cSLEDAI-2K; *p* = 0.05), whereas acquired organ damage, as assessed by the Systemic Lupus International Collaborating Clinics/ACR damage index (SDI), was lower (*p* = 0.006) in the anti-U1-RNP-positive individuals with SLE (Table 1). In addition, SLE samples that contained anti-U1-RNP antibodies had lower eosinophil counts (*p* = 0.0005), lower lymphocyte counts (*p* = 0.006), lower monocyte counts (*p* = 0.03) and lower concentrations of complement protein (C) 3 (*p* = 0.03) compared to samples from the anti-U1-RNP-negative group.

### 2.2. Anti-U1-RNP and Anti-RNP70 Antibodies (FEIA) in SLE, MCTD, pSS and HBDs

In total, 34/114 (30%) patients with SLE were found to be anti-U1-RNP positive, and 21/114 (18%) were anti-RNP70 positive, according to the FEIAs. As illustrated in the Venn diagrams, all the anti-RNP70-positive samples were also found to be positive for anti-U1-RNP antibodies using the FEIAs (Figure 2). Overall, 32/53 (60%) SLE sera that had previously tested positive for anti-U1-RNP antibodies in the clinical routine (ALBIA and/or EUROLINE) could be confirmed by the FEIAs. No significant associations were found between the presence of anti-U1-RNP or anti-RNP70 antibodies and secondary Sjögren’s syndrome (sSS) in individuals with SLE.

In all, 10/12 (83%) sera from patients with MCTD who previously had tested positive for anti-U1-RNP antibodies (ALBIA and/or EUROLINE) were confirmed to be anti-U1-RNP-positive, and 9/12 (75%) were determined to be anti-RNP70 positive with the FEIAs.

All of the anti-RNP70-positive sera (*n* = 21) were anti-U1-RNP positive according to the FEIAs, and one of these was judged to be anti-U1-RNP negative in the clinical routine. In addition, 59 (97%) out of the 61 SLE sera that had been deemed to be anti-U1-RNP negative with the ALBIA and/or EUROLINE were also negative according to the FEIAs. One serum was determined to be positive with the FEIAs for both anti-U1-RNP and anti-RNP70 antibodies, and the other was only positive for anti-U1-RNP.

Among the individuals with pSS (*n* = 54), one was positive and carried antibodies against both U1-RNP and RNP70 (1.9%) and had at the time of sampling also been anti-U1-RNP positive according to the EUROLINE immunoblot. Among the 128 HBDs, none tested positive for U1-RNP or RNP70 in the FEIAs.

In total, the concordance was 90% between the previous results for anti-U1-RNP antibodies detected in the clinical routine with the ALBIA and/or the EUROLINE immunoblot and the study results obtained with the FEIAs (229/254).

### 2.3. The Added Value of Analyzing Anti-RNP70 Antibodies in Cases of SLE

We stratified the patients with SLE who were positive for anti-U1-RNP antibodies (FEIA, *n* = 34) into those with (*n* = 21) or without (*n* = 13) concomitant anti-RNP70 antibodies and found no differences regarding clinical characteristics (age, disease duration, ACR criteria, cSLEDAI-2K, SDI and RP) or laboratory variables (blood cell counts, C3 and C4) between the groups.

A comparison of the consistently anti-U1-RNP-negative samples (*n* = 59) with the anti-RNP70-positive (*n* = 21) SLE samples showed that the anti-RNP70-positive SLE subgroup had a lower age (*p* = 0.001), a shorter disease duration (*p* < 0.0001) and a lower SDI (*p* = 0.008), while the fulfillment of the hematologic disorder criterion (*p* = 0.007) and the presence of RP (*p* = 0.003) were more common compared to the anti-U1-RNP-negative group. In addition, patients with anti-RNP70 antibodies (according to the FEIAs) had lower hemoglobin concentrations (*p* = 0.04) and lower leukocyte (*p* = 0.04), basophil (*p* = 0.047), monocyte (*p* = 0.049), eosinophil (*p* = 0.0005) and lymphocyte (*p* = 0.008) counts (Table 2).

### 2.4. Importance of Anti-U1-RNP and Anti-RNP70 Antibody Levels (FEIA)

Among the patients with SLE, we observed that serum samples that contained antibodies against U1-RNP in combination with RNP70 (FEIA) showed significantly higher levels of anti-U1-RNP antibodies than those that were exclusively anti-U1-RNP positive (*p* < 0.00001). Otherwise, no statistically significant correlations were observed between the anti-U1-RNP or anti-RNP70 antibody levels and the continuous variables. For SLE disease activity assessed by cSLEDAI-2K, neither the anti-U1-RNP (*p* = 0.41) nor the anti-RNP70 (*p* = 0.59) antibody levels showed significant correlations.

Finally, the antibody levels of patients with SLE and MCTD were compared. Whereas the anti-U1-RNP antibody levels were similar between antibody-positive patients with SLE and MCTD, there was a non-significant trend of higher anti-RNP70 antibody levels among subjects with MCTD (Appendix A).

## 3. Discussion

In this cross-sectional study, we examine the presence of anti-U1-RNP and anti-RNP70 antibodies in diseases that have similar clinical phenotypes and immunopathologies [3]. Our results show that anti-RNP70 antibodies are often but not exclusively found in MCTD but are rarely detected in pSS cases and HBDs. Among patients with SLE, we observed no significant differences in clinical or laboratory parameters between anti-U1-RNP-positive individuals who carried or lacked anti-RNP70 antibodies. In line with previous reports, the presence of anti-U1-RNP antibodies was associated with a clinical phenotype that resembles MCTD with hematologic involvement and less damage accrual. However, based on our data, the clinical value of subtyping anti-RNP70 antibodies in anti-U1-RNP-positive sera from SLE patients appears to be limited.

The activation of the type I interferon (IFN) system is ubiquitous in cases of SLE and pSS, and anti-U1-RNP-containing immune complexes (ICs) have the ability to elicit type I IFNs via Toll-like receptor 7 [20]. As the analysis of anti-RNP70 antibodies has been claimed to have advantages in terms of diagnosing MCTD [15,16,17,18], we evaluated this further using relevant comparators for anti-U1-RNP antibodies that are usually available in routine ANA screening.

By using the EliA™ assays at the accredited Clinical Immunology Laboratory in Linköping, we observed that the presence of anti-U1-RNP antibodies concomitant with anti-RNP70 antibodies was almost exclusively observed for samples that originated from patients with SLE or MCTD. Based on the knowledge that patients with MCTD have higher levels of anti-U1-RNP antibodies, compared to patients with SLE, it is interesting to note that combined positivity (anti-U1-RNP plus anti-RNP70) results in overall higher anti-U1-RNP antibody levels [12]. This may indicate a broader immune reaction against the U1-snRNP complex in individuals with an MCTD-like disease phenotype. However, in neither the SLE group nor the MCTD group did we find any patient who was positive for anti-RNP70 without being positive for anti-U1-RNP antibodies (FEIA).

The level of agreement between anti-U1-RNP in clinical routine testing and the FEIAs was not perfect (concordance of 90%), indicating that the assays have different advantages in terms of analytical sensitivity and specificity. However, the prevalence rates of these autoantibodies in the SLE and MCTD cohorts were in agreement with those reported by others [6]. Only one patient with pSS (1.9%) was anti-U1-RNP/RNP70 positive, which is in the same range as that reported by Abbara et al., in which 21/467 patients with pSS were anti-U1-RNP positive [21]. Not surprisingly, we found that patients with MCTD were most frequently positive for anti-U1-RNP (83%) and anti-RNP70 (75%) antibodies. Regarding SLE, we obtained samples from the longitudinal quality and research register KLURING (Swedish acronym for *Clinical Lupus Register In North-eastern Gothia*), with individuals whose blood tests and assessments of disease activity and organ damage were available from the time of sampling. For the current study, we chose to include samples from patients who tested positive or negative for anti-U1-RNP antibodies in the clinical routine.

In the case of SLE, we observed that anti-RNP70 positivity compared to anti-U1-RNP negativity was associated with multiple hematologic aberrations, as reflected by more of the subjects fulfilling the hematologic disorder ACR criterion and having lower counts of leukocytes, lymphocytes, eosinophils, monocytes and basophils, as well as lower hemoglobin concentrations. In addition, the patients with anti-RNP70 antibodies appeared to have accrued less organ damage and were more likely to suffer from RP than the anti-U1-RNP-negative subjects. The corresponding comparison between anti-U1-RNP-positive and -negative patients with SLE using our routine assays showed lower levels of C3 and less accrued organ damage in the patients who were anti-U1-RNP positive, with these individuals more commonly suffering from RP and fulfilling the hematologic disorder ACR criterion, as well as exhibiting lower counts of eosinophils, lymphocytes and monocytes.

When we compared the anti-U1-RNP-positive (FEIA) patients with or without concomitant anti-RNP70 antibodies (FEIA), we did not detect any differences regarding the study variables. This indicates that analyzing anti-RNP70 antibodies does not add much clinical value compared to analyzing anti-U1-RNP antibodies with multiple specificities for RNP A, C and the 70 kDa subunit. However, the number of patients examined might have been too low to detect an actual difference. Currently, efforts are being made to screen for autoantibodies using an in vitro proteome, which can reveal the “autoantibody landscape” of human subjects and may identify novel clinical biomarkers. These sophisticated assays, which have the potential to differentiate individuals with autoimmune diseases from both patients with cancer and healthy subjects, may soon be clinically available [22].

The biological functions of anti-U1-RNP antibodies have recently been studied in detail by Hubbard et al. [23]. A complex relationship between the presence of these autoantibodies and the IFN gene signature in cases of SLE has been observed, with an impact of ancestry in African and European patients. Furthermore, in the study conducted by Hubbard et al. [23], clear-cut associations were found between anti-dsDNA antibody levels and C3, as well as C4 levels; however, such an association was not observed for anti-U1-RNP antibodies, suggesting that anti-U1-RNP ICs drive the IFN gene signature without complement activation. In our hands, only anti-U1-RNP antibodies (and not anti-RNP70 antibodies) were associated with lower C3 concentrations.

The limitations of this study include the low number of patients with MCTD, although this disease is indeed rare. We acknowledge that the lack of pre-analytical statistical power calculation is a limitation. The study had a cross-sectional design, and no longitudinal analyses of antibody specificities were conducted, which constitutes a potential limitation. However, follow-up data from similar studies indicate that the anti-U1-RNP and anti-RNP70 antibodies remain stable over time [9]. There are several strengths of the study, e.g., the relevant and well-characterized disease controls and the large group of HBDs. The Swedish healthcare system is public and taxpayer funded and offers universal access, which excludes the risk of selection bias. Another advantage was that all the antibody analyses were performed at the same time by an accredited laboratory.

To summarize, anti-U1-RNP and anti-RNP70 antibodies detected by FEIAs were found almost exclusively in MCTD and SLE cases compared to pSS cases and HBDs. In our hands, based on the limited clinical associations achieved, separate analyses of anti-RNP70 and anti-U1-RNP antibodies in cases of SLE cannot be recommended. Anti-RNP70 antibodies clearly do not facilitate the differentiation of SLE from MCTD. However, in subjects with SLE, both anti-RNP antibody specificities were associated with multiple hematologic aberrations, the presence of RP and less organ damage. Our findings provide support for the idea that patients with SLE who carry anti-U1-RNP and/or anti-RNP70 antibodies exhibit a clinical phenotype that resembles MCTD with hematologic involvement and an overall less severe disease [24].

## 4. Patients and Methods

### 4.1. Study Population and Data Collection

In the current study, we included serum samples that were collected from 114 patients who were diagnosed with SLE, all of whom met the 1982 ACR and/or the 2012 Systemic Lupus International Collaborating Clinics (SLICC) classification criteria and had taken part in the prospective follow-up program KLURING at the Rheumatology Clinic, Linköping University Hospital [25]. None of the 114 patients fulfilled the criteria for pSS, MCTD or rheumatoid arthritis. One patient with SLE had concomitant autoimmune hepatitis, and 21/114 (18%) of the patients with SLE also had sSS.

Of the 114 serum samples, 53 (46%) had previously tested positive for anti-U1-RNP antibodies, and 61 (54%) had tested negative for anti-U1-RNP antibodies, in the clinical routine using either ALBIA (FIDIS™ Connective profile; Theradiag, Croissy-Beaubourg, France), EUROLINE Anti-ENA ProfilePlus-1 (IgG) immunoblot (EUROIMMUN AG, Lübeck, Germany) or both assays [4]. These results were referred to as “clinical routine”, in contrast to the FEIA results, as illustrated in the flowchart (Figure 1).

Clinical routine analyses at sampling were performed at the Clinical Chemistry Laboratory (Linköping University Hospital). Plasma complement proteins C3 and C4 were measured using immunoturbidimetry with Roche Cobas^®^ c502 (Roche Diagnostics Scandinavia AB, Solna, Sweden). SLE disease activity was assessed using cSLEDAI-2K, which, in contrast to SLEDAI-2K, excludes items for low complement levels and positive anti-dsDNA antibodies [26]. Irreversible organ damage, which was required to have been persistent for ≥6 months, was recorded at the time of blood sampling using the SDI, which encompasses damage in 12 defined organ systems [26]. The clinical characteristics of the participating subjects with SLE are detailed in Table 1.

In addition, we included sera from 54 subjects with pSS who met the American–European consensus criteria (52 women, 2 men; mean age, 62 years; range, 25–83 years), and 12 women (mean age, 34 years; range, 14–54 years) who fulfilled the proposed diagnostic criteria for MCTD (including a historically positive ALBIA/EUROLINE immunoblot anti-U1-RNP screening test, which had been confirmed by immunodiffusion technique) [12,27]. Serum samples from 128 HBDs (114 women, 14 men; mean age, 43 years; range, 20–67 years) served as controls. Of these 128 sera, 24 (19%) were selected due to testing positive for ANAs using IIF microscopy performed at the Clinical Immunology Laboratory, Linköping University Hospital, whereas the remaining 104 sera were ANA negative. None of the ANA-positive HBDs tested positive for anti-U1-RNP antibodies by ALBIA (Figure 1).

### 4.2. Immunoassays

The fluorescence enzyme immunoassays EliA™ U1-RNP and EliA™ RNP70 were used to detect immunoglobulin (Ig)G anti-U1-RNP and anti-RNP70 at the accredited Clinical Immunology Laboratory, Linköping University Hospital, Sweden (Phadia 250; Thermo-Fisher Scientific, Phadia AB, Uppsala, Sweden), in accordance with the manufacturer’s instructions. Briefly, the U1-RNP wells were coated with human recombinant RNP (RNP70, A, C) proteins, whereas the RNP70 wells were coated with the human recombinant RNP70 antigen. The wells were incubated with patient sera followed by a washing step and the addition of enzyme/β-galactosidase-conjugated anti-human-IgG antibodies. After incubation and washing, a substrate for β-galactosidase was added, and the reaction was terminated with a stop solution. The fluorescence was quantified using a calibration curve, and the results are reported in U/mL. The cutoffs provided by the manufacturer (Thermo-Fisher) were applied according to the following: for U1-RNP, negative (<5 U/mL), equivocal (5–10 U/mL) and positive (>10 U/mL); for RNP70, negative (<7 U/mL), equivocal (7–10 U/mL) and positive (>10 U/mL). Equivocal results were considered to be negative.

### 4.3. Statistics

For comparisons of U1-RNP/RNP70 antibody levels between the groups, the Mann–Whitney *U*-test was used. Associations between U1-RNP/RNP70 antibody positivity (categorical variable) and clinical manifestations or organ damage were examined with the χ^2^ test or Fisher’s exact test when appropriate (*n* ≤ 5). Correlation analyses were performed using Spearman’s rank correlation. A two-sided *p*-value <0.05 was considered statistically significant. Concordance between methods was defined as the sum of the double-positive samples and double-negative samples divided by the total number of samples and multiplied by 100.

### 4.4. Ethical Considerations

Oral and written informed consent was obtained from all patients and healthy controls. The study was conducted according to the Declaration of Helsinki, and the study protocol was approved by the Regional Ethics Boards (Linköping M75–08/2008; Linköping 2017/474–31; the Swedish Ethical Review Authority 2020–03287).

## Figures and Tables

**Figure 1 ijms-24-10398-f001:**
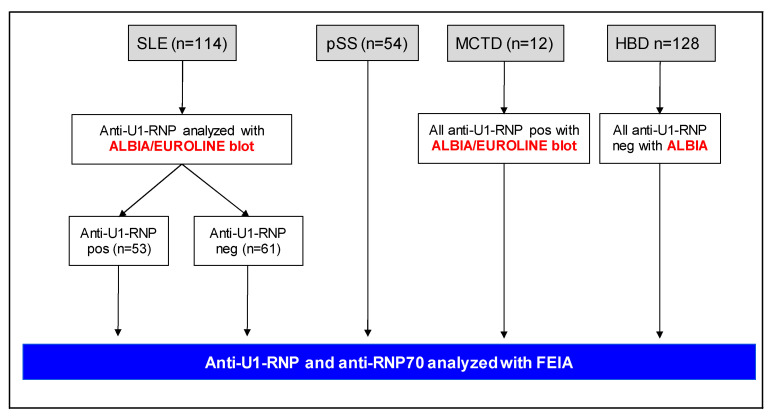
Flowchart illustrating the four included cohorts that were eligible for analyses with the fluoroenzymatic immunoassay (FEIA): systemic lupus erythematosus (SLE); primary Sjögren’s syndrome (pSS); mixed connective tissue disease (MCTD); and healthy blood donors (HBD). Addressable Laser Bead ImmunoAssay (ALBIA) and/or EUROLINE immunoblot refer to the clinical routine.

**Figure 2 ijms-24-10398-f002:**
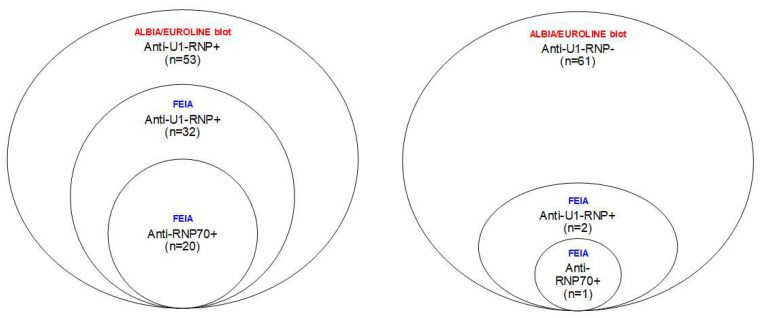
Venn diagrams of the samples from the two subgroups of SLE patients, showing the results of the EUROLINE immunoblot and/or the addressable laser bead immunoassay (ALBIA) used in the clinical laboratory routine at the time of sampling and the results obtained with the fluoroenzymatic immunoassays (FEIA). The left panel depicts the samples (*n* = 53) that were positive, and the right panel shows the samples (*n* = 61) that tested negative for anti-U1-RNP antibodies in the clinical routine. All of the samples that tested positive for anti-RNP70 were also positive for anti-U1-RNP with FEIAs.

**Table 1 ijms-24-10398-t001:** Characteristics of the included patients with SLE categorized according to the anti-U1-RNP antibody results obtained in the clinical routine (Addressable Laser Bead ImmunoAssay (ALBIA) and/or EUROLINE immunoblot).

	SLE: Anti-U1-RNP Positive (*n* = 53)	SLE: Anti-U1-RNP Negative (*n* = 61)	*p*-Value
**Background variables**			
Females, *n* (%)	49 (92.5)	56 (91.8)	n.s.
Age in years, mean (range)	46 (23–79)	57 (27–87)	<0.0001
SLE duration in years, mean (range)	10 (0–47)	23 (9–48)	<0.0001
Caucasian race/ethnicity, *n* (%)	46 (86.8)	56 (91.8)	n.s.
cSLEDAI-2K, mean (range)	1.9 (0–20)	0.77 (0–12)	n.s.
Raynaud’s phenomenon, *n* (%)	26 (49.0)	18 (29.5)	0.03
SDI, mean (range)	0.9 (0–7)	1.7 (0–7)	0.006
**Fulfilled classification criteria** (1982 ACR definitions)			
Malar rash, *n* (%)	23 (43.4)	32 (52.5)	n.s.
Discoid lupus, *n* (%)	9 (17)	6 (9.8)	n.s.
Photosensitivity, *n* (%)	30 (56.6)	30 (49.2)	n.s.
Oral ulcers, *n* (%)	9 (17)	6 (9.8)	n.s.
Arthritis, *n* (%)	36 (67.9)	46 (75.4)	n.s.
Serositis, *n* (%)	13 (24.5)	25 (41)	n.s.
Renal disorder, *n* (%)	17 (32.1)	19 (31.1)	n.s.
Neurologic disorder, *n* (%)	3 (5.7)	5 (8.2)	n.s.
Hematologic disorder, *n* (%)	38 (71.7)	32 (52.5)	0.04
Immunologic disorder, *n* (%)	33 (62.3)	31 (50.8)	n.s.
IIF-ANA, *n* (%)	53 (100)	61 (100)	n.s.
**Medical treatment** (ongoing)			
Hydroxychloroquine, *n* (%)	38 (72)	45 (74)	n.s.
^1^ Immunosuppressants, *n* (%)	23 (43)	28 (46)	n.s.
Prednisolone, daily dose (mg)	4.5	2.6	0.03
**Laboratory variables at sampling occasion**, mean (range)			
Hemoglobin concentration, g/L	128 (52–177)	133 (103–171)	n.s.
Platelet count, ×10^9^/L	230 (102–485)	244 (132–500)	n.s.
Leukocyte count, ×10^9^/L	5.7 (2.9–15.8)	6.4 (3.0–14.5)	n.s.
Basophil count, ×10^9^/L	0.03 (0.01–0.15)	0.04 (0.01–0.16)	n.s.
Eosinophil count, ×10^9^/L	0.06 (0.01–0.31)	0.13 (0.01–0.77)	0.0005
Lymphocyte count, ×10^9^/L	1.2 (0.4–3.7)	1.6 (0.4–4.5)	0.006
Monocyte count, ×10^9^/L	0.41 (0.24–0.41)	0.50 (0.19–1.46)	0.03
C3, g/L	0.96 (0.36–1.40)	1.05 (0.63–1.60)	0.03
C4, g/L	0.16 (0.04–0.30)	0.17 (0.05–0.49)	n.s.

cSLEDAI-2K, Clinical SLE disease activity index 2000; IIF-ANA, antinuclear antibodies detected by indirect immunofluorescence microscopy; n.s., not significant; SDI, Systemic Lupus International Collaborating Clinics/American College of Rheumatology damage index. ^1^ Immunosuppressive therapy included mycophenolate mofetil, methotrexate, leflunomide, azathioprine, sirolimus, bortezomib, rituximab and belimumab.

**Table 2 ijms-24-10398-t002:** Comparisons of variables between the anti-RNP70-positive and anti-U1-RNP-negative subjects with SLE, based on results from the fluoroenzymatic immunoassays (FEIA).

	SLE: Anti-RNP70 Positive (*n* = 21)	SLE: Anti-U1-RNP Negative (*n* = 59)	*p*-Value
**Background variables**			
Females, *n* (%)	20 (95.2)	54 (91.5)	n.s.
Age in years, mean (range)	44 (26–78)	58 (27–87)	0.001
SLE duration in years, mean (range)	11 (1–30)	23 (9–48)	<0.0001
Caucasian race/ethnicity, *n* (%)	17 (81.0)	54 (91.5)	n.s.
cSLEDAI-2K, mean (range)	1.4 (0–4)	0.8 (0–12)	n.s.
Raynaud’s phenomenon, *n* (%)	14 (66.7)	17 (29.8)	0.003
SDI, mean (range)	0.7 (0–4)	1.8 (0–7)	0.008
**Fulfilled classification criteria** (1982 ACR definitions)			
Malar rash, *n* (%)	9 (42.9)	30 (50.8)	n.s.
Discoid lupus, *n* (%)	2 (9.5)	6 (10.6)	n.s.
Photosensitivity, *n* (%)	9 (42.9)	28 (47.5)	n.s.
Oral ulcers, *n* (%)	5 (23.8)	6 (10.2)	n.s.
Arthritis, *n* (%)	16 (76.2)	44 (74.6)	n.s.
Serositis, *n* (%)	5 (23.8)	24 (40.7)	n.s.
Renal disorder, *n* (%)	7 (33.3)	19 (32.2)	n.s.
Neurologic disorder, *n* (%)	1 (4.8)	5 (8.5)	n.s.
Hematologic disorder, *n* (%)	18 (85.7)	31 (52.5)	0.007
Immunologic disorder, *n* (%)	14 (66.7)	30 (50.8)	n.s.
IIF-ANA, *n* (%)	21 (100)	59 (100)	n.s.
**Laboratory variables at sampling occasion**, mean (range)			
Hemoglobin concentration, g/L	124 (52–164)	132 (103–171)	0.04
Platelet count, ×10^9^/L	230 (102–349)	243 (132–500)	n.s.
Leukocyte count, ×10^9^/L	5.2 (2.9–15.8)	6.4 (3.0–14.5)	0.04
Basophil count, ×10^9^/L	0.03 (0.01–0.10)	0.04 (0.01–0.16)	<0.05
Eosinophil count, ×10^9^/L	0.09 (0.01–0.31)	0.13 (0.01–0.77)	0.0005
Lymphocyte count, ×10^9^/L	1.1 (0.6–2.6)	1.6 (0.4–4.5)	0.008
Monocyte count, ×10^9^/L	0.39 (0.21–1.46)	0.50 (0.19–1.46)	<0.05
C3, g/L	0.96 (0.47–1.40)	1.06 (0.63–1.60)	n.s.
C4, g/L	0.15 (0.08–0.28)	0.17 (0.05–0.49)	n.s.

cSLEDAI-2K, Clinical SLE disease activity index 2000; C, complement protein; IIF-ANA, antinuclear antibodies detected by indirect immunofluorescence microscopy; n.s., not significant; SDI, Systemic Lupus International Collaborating Clinics/American College of Rheumatology damage index.

## Data Availability

All datasets generated for this study are included in the article.

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
