# Peer review of "Doubtful Clinical Value of Subtyping Anti-U1-RNP Antibodies Regarding the RNP-70 kDa Antigen in Sera of Patients with Systemic Lupus Erythematosus"

_ijms, 2023, doi:10.3390/ijms241210398_

Round 1

Reviewer 1 Report

 Limited clinical value of subtyping Anti-U1-RNP antibodies 2 regarding RNP-70kDa in Systemic Lupus Erythematosus

Comments

15…diagnostics and prognostics…diagnosis and prognosis

18.. 21/114 (18%) …it is not preferred to start a sentence with a number.

21.. were younger, showed lower C3, had lower blood cell....it is better to include numbers when talk about results.

The abstract should be more concise and targeted, and the conclusion should reflect the results.

It is better to add the research gap before the aim of the study at the last part of the introduction.

The authors should determine how they calculated the sample size.

103.. the manuscript should be revised for English editing and grammar correction .The authors used numbers multiple times at the start

166.. This cross-sectional study included several objectives …the English language need to be more scientific

166..the first paragraph of discussion should include a brief summary of the results

Add the strength and recommendations at the end of discussion 

Moderate editing of English language required

Author Response

Comments and Suggestions for Authors

Limited clinical value of subtyping Anti-U1-RNP antibodies 2 regarding RNP-70kDa in Systemic Lupus Erythematosus

Response: Thank you for the careful review of our manuscript.

Comments

15…diagnostics and prognostics…diagnosis and prognosis

Response: Thank you. Now corrected.

18.. 21/114 (18%) …it is not preferred to start a sentence with a number.

Response: Thank you. Now corrected.

21.. were younger, showed lower C3, had lower blood cell....it is better to include numbers when talk about results.

Response: Thank you. We included p-values in the Abstract of the revised manuscript.

The abstract should be more concise and targeted, and the conclusion should reflect the results.

Response: We have revised the Abstract in order to make it more concise and targeted.

It is better to add the research gap before the aim of the study at the last part of the introduction.

Response: Good suggestion. The last part of the Introduction has been revised accordingly.

The authors should determine how they calculated the sample size.

Response: In fact, no statistical power calculation was performed. Instead, all anti-U1-RNP positive samples from the SLE cohort “Clinical Lupus Register In North-eastern Gothia” were used and compared to a similar amount of anti-U1-RNP negative cases. This information has been added (page 8, line 222-226). We agree in that the lack of pre-analytical power calculation is a limitation, which we also acknowledge (page 9, line 260-261).

103.. the manuscript should be revised for English editing and grammar correction. The authors used numbers multiple times at the start

Response: Thank you. The revised manuscript has now been thoroughly reviewed by a native English-speaking person.

166.. This cross-sectional study included several objectives …the English language need to be more scientific

Response: See above. The revised manuscript has now been thoroughly reviewed by a native English-speaking person with a scientific background in biomedicine.

166..the first paragraph of discussion should include a brief summary of the results

Response: Good suggestion. We have re-written the first part of the Discussion.

Add the strength and recommendations at the end of discussion

Response: Thank you. The last part of the Discussion has been extensively re-written to include a clear recommendation based on our findings. Strengths and limitations of the study are discussed in the section above this.

Reviewer 2 Report

In this article, Ahmad et al analyzed anti-U1-RNP and anti-RNP70 antibodies with commercial immunoassays in sera from patients with SLE, pSS, MCTD and healthy blood donors (HBD). Anti-U1-RNP positive subjects with SLE were younger, showed lower C3, had lower blood cell counts and had accrued less organ damage than anti-U1-RNP negatives. However, they observed no clinical or laboratory significant differences between anti-U1-RNP positives ± anti-RNP70 antibodies in SLE. They concluded that anti-RNP70 antibodies were not exclusively found in MCTD. In SLE, anti-U1-RNP associates with a clinical phenotype resembling MCTD with hematological involvement and less damage accrual but, based on their results, the clinical value of subtyping anti-RNP70 in anti-U1-RNP antibody positive sera appears to be of limited value. This is a very interesting report, however, because of the deviation from the previous report, I have the following questions regarding the background factors and other aspects that are not mentioned.

major concerns)

1) It is known that these and other antibody titers can be affected by prior therapy, but what was the prior therapy? I have not seen this stated, so please state.

2) The following is a previous report on the clinical significance of anti-RNP70 antibodies.

(Masuyuki Nawata, et al. Clinical Significance of Anti─U1 RNP Antibodies Recognizing The Conformation Structure on U1 RNA/70─kd Protein Complex in Patients with Mixed Connective Tissue Disease. Juntendo Medical Journal, 2011. 57(5), 477-487. https://www.jstage.jst.go.jp/article/pjmj/57/5/57_477/_pdf/-char/ja) In this literature, anti-RNP70 antibodies are reported to be significantly associated with the diagnosis of CNS lupus and SLE . The major difference between this article and the previous report is that the kit is only used in this study, and U1-RNA is not included in this study. The experimental system in this paper, although simple, is considered to be insufficient as an experimental system, and additional experiments are needed.

3) In relation to 2) above, an array has recently been developed as a more detailed testing method than Euroline, which detects antibodies using a protein that retains its three-dimensional structure (Matsuda KM, Yoshizaki A, Yamaguchi K, Fukuda E, Okumura T, Ogawa K, Ono C, Norimatsu Y, Kotani H, Hisamoto T, Kawanabe R, Kuzumi A, Fukasawa T, Ebata S, Miyagawa T, Yoshizaki-Ogawa A, Goshima N, Sato S. Autoantibody Landscape Revealed by Wet Protein Array: Sum of Autoantibody Levels Reflects Disease Status. Front Immunol. 2022 May 4;13:893086. doi: 10.3389/fimmu.2022.893086. PMID: 35603173; PMCID: PMC9114879.) It is considered that it is better to reexamine using that method in order to obtain more accurate data.

4) In Table 2, since all of the patients positive for anti-RNP70 antibody are positive for anti-U1-RNP antibody, it is not surprising that a comparison with and without anti-RNP70 antibody would yield the same results as with and without anti-U1-RNP antibody. If a detailed study is conducted, would it be possible to understand the significance of anti-RNP70 antibody detection by comparing patients with and without anti-RNP70 antibody among those with anti-U1-RNP antibody-positive patients? It is also possible that among patients positive for anti-RNP70 antibody, their titer may correlate with their clinical score. A detailed study is needed to avoid misleading future studies.

I had no problems understanding the English content.

Author Response

Comments and Suggestions for Authors

In this article, Ahmad et al analyzed anti-U1-RNP and anti-RNP70 antibodies with commercial immunoassays in sera from patients with SLE, pSS, MCTD and healthy blood donors (HBD). Anti-U1-RNP positive subjects with SLE were younger, showed lower C3, had lower blood cell counts and had accrued less organ damage than anti-U1-RNP negatives. However, they observed no clinical or laboratory significant differences between anti-U1-RNP positives ± anti-RNP70 antibodies in SLE. They concluded that anti-RNP70 antibodies were not exclusively found in MCTD. In SLE, anti-U1-RNP associates with a clinical phenotype resembling MCTD with hematological involvement and less damage accrual but, based on their results, the clinical value of subtyping anti-RNP70 in anti-U1-RNP antibody positive sera appears to be of limited value. This is a very interesting report, however, because of the deviation from the previous report, I have the following questions regarding the background factors and other aspects that are not mentioned.

Response: Thank you for the careful review of our manuscript.

major concerns

1) It is known that these and other antibody titers can be affected by prior therapy, but what was the prior therapy? I have not seen this stated, so please state.

Response: This is a relevant question. In contrast to anti-dsDNA and anti-C1q antibodies, anti-U1-RNP titers usually do not follow SLE disease activity. However, it cannot be excluded that prior medication may have had an impact on antibody levels. We have studied the specific effects of anti-U1-RNP titers in detail over time and in relation to medication in a previous paper (Frodlund M, et al. Clin Exp Immunol 2020, PMID: 31778219) but it was outside the scope of this project, and we definitely did not have the statistical power to evaluate any effect of immunosuppressive treatment on the antibody titers. Nevertheless, according to your suggestion, we have included new information in Table 1 regarding ongoing therapy (page 4-5).

2) The following is a previous report on the clinical significance of anti-RNP70 antibodies.

(Masuyuki Nawata, et al. Clinical Significance of Anti─U1 RNP Antibodies Recognizing The Conformation Structure on U1 RNA/70─kd Protein Complex in Patients with Mixed Connective Tissue Disease. Juntendo Medical Journal, 2011. 57(5), 477-487. https://www.jstage.jst.go.jp/article/pjmj/57/5/57_477/_pdf/-char/ja) In this literature, anti-RNP70 antibodies are reported to be significantly associated with the diagnosis of CNS lupus and SLE . The major difference between this article and the previous report is that the kit is only used in this study, and U1-RNA is not included in this study. The experimental system in this paper, although simple, is considered to be insufficient as an experimental system, and additional experiments are needed.

Response: Thank you. We now refer to this relevant study in the revised version of our manuscript. However, the aim of our manuscript was different as we aimed to evaluate two commercially available immunoassays.

3) In relation to 2) above, an array has recently been developed as a more detailed testing method than Euroline, which detects antibodies using a protein that retains its three-dimensional structure (Matsuda KM, Yoshizaki A, Yamaguchi K, Fukuda E, Okumura T, Ogawa K, Ono C, Norimatsu Y, Kotani H, Hisamoto T, Kawanabe R, Kuzumi A, Fukasawa T, Ebata S, Miyagawa T, Yoshizaki-Ogawa A, Goshima N, Sato S. Autoantibody Landscape Revealed by Wet Protein Array: Sum of Autoantibody Levels Reflects Disease Status. Front Immunol. 2022 May 4;13:893086. doi: 10.3389/fimmu.2022.893086. PMID: 35603173; PMCID: PMC9114879.) It is considered that it is better to reexamine using that method in order to obtain more accurate data.

Response: Thank you for highlighting this paper, which we were not aware of. We have added this relevant information to the Discussion (page 9, line 244-249). However, the aim of our project was slightly different as we aimed to evaluate two currently available commercial immunoassays that are using recombinant antigens.

4) In Table 2, since all of the patients positive for anti-RNP70 antibody are positive for anti-U1-RNP antibody, it is not surprising that a comparison with and without anti-RNP70 antibody would yield the same results as with and without anti-U1-RNP antibody. If a detailed study is conducted, would it be possible to understand the significance of anti-RNP70 antibody detection by comparing patients with and without anti-RNP70 antibody among those with anti-U1-RNP antibody-positive patients? It is also possible that among patients positive for anti-RNP70 antibody, their titer may correlate with their clinical score. A detailed study is needed to avoid misleading future studies.

Response: Yes, as expected, the overlap between anti-U1-RNP and anti-RNP70 was large but not 100%. The main focus of this paper was to evaluate the added value of subtyping anti-U1-RNP antibody positive SLE sera regarding anti-RNP70. The size of the study population was not that large and we acknowledge this as a limitation (page 9, line 259-263). According to your suggestion, we performed additional analysis on the association between anti-RNP70 antibody levels and disease activity assessed by clinical SLEDAI-2K (page 7, line 178-179).

Comments on the Quality of English Language

I had no problems understanding the English content.

Reviewer 3 Report

The Authors presented an interesting study on the limited clinical value of subtyping Anti-U1-RNP antibodies. Pointing out problems concerning the diagnosis of autoimmune diseases. The paper is clearly written; however, I have a few comments. The following are listed:

1. In the title, the authors indicate the usefulness of the antibodies in the diagnosis of Systemic Lupus Erythematosus, but already in the abstract they indicate the extension of the application to other disorders. There is some inconsistency here, particularly as the results do not only refer to SLE.

2. The paper should also include information on the similarities in the clinical course and difficulties in diagnosing the disorders studied. I also miss information on the symptoms of the disorders. It would greatly help to indicate the validity of the search for unambiguous diagnostic tests.

3 Line 21 - all abbreviations for the conditions discussed are explained. Only complement protein is described as C3 without a word of explanation.

4. The paper also lacks a Table summarising the numerical data described in section 2.2.

Author Response

Comments and Suggestions for Authors

The Authors presented an interesting study on the limited clinical value of subtyping Anti-U1-RNP antibodies. Pointing out problems concerning the diagnosis of autoimmune diseases. The paper is clearly written; however, I have a few comments.

The following are listed:

Response: Thank you for the careful review of our manuscript.

  1. In the title, the authors indicate the usefulness of the antibodies in the diagnosis of Systemic Lupus Erythematosus, but already in the abstract they indicate the extension of the application to other disorders. There is some inconsistency here, particularly as the results do not only refer to SLE.

Response: This is a relevant question. We had detailed clinical information available only for the patients in the SLE cohort. That is why we focused on the presence of anti-U1-RNP and anti-RNP70 antibodies in relation to clinical manifestations and laboratory variables in SLE only. The MCTD, pSS and HBD samples were used as suitable comparators. We also discuss similarities between SLE, pSS and MCTD with regard to immunopathology and the type I interferon system (page 8, line 197-201).

  1. The paper should also include information on the similarities in the clinical course and difficulties in diagnosing the disorders studied. I also miss information on the symptoms of the disorders. It would greatly help to indicate the validity of the search for unambiguous diagnostic tests.

Response: Thank you. We have added relevant text and new references in the Introduction about this.

3 Line 21 - all abbreviations for the conditions discussed are explained. Only complement protein is described as C3 without a word of explanation.

Response: Thank you. This has been corrected.

  1. The paper also lacks a Table summarising the numerical data described in section 2.2.

Response: Thank you. As this manuscript is written in a “Brief Report” format, we decided to avoid inclusion of additional tables and prefer to present the data in section 2.2. entirely in text.

Reviewer 4 Report

In my opinion, article entitled "Limited clinical value of subtyping Anti-U1-RNP antibodies regarding RNP-70kDa in Systemic Lupus Erythematosus" is well prepared (from the technical side too) and includes sufficient material as brief report. It could be interesting for the readers of this Journal. Eventually, figures could visulize more details from experiments.

Author Response

Comments and Suggestions for Authors

In my opinion, article entitled "Limited clinical value of subtyping Anti-U1-RNP antibodies regarding RNP-70kDa in Systemic Lupus Erythematosus" is well prepared (from the technical side too) and includes sufficient material as brief report. It could be interesting for the readers of this Journal. Eventually, figures could visulize more details from experiments.

Response: Thank you for the careful review of our manuscript. As this manuscript is written in a “Brief Report” format, we decided to avoid inclusion of additional tables and figures.

Round 2

Reviewer 1 Report

The manuscript has improved to a great extent 

Minor editing of English language required

Reviewer 2 Report

Authors replied to our questions adequately. There are no additional questions.